# DiscoveryBench: Towards Data-Driven Discovery with Large Language Models

**Bodhisattwa Prasad Majumder**[*α]    **Harshit Surana**[*αβ]    **Dhruv Agarwal**[*γ]
**Bhavana Dalvi Mishra**[*α]    **Abhijeetsingh Meena**[β]    **Aryan Prakhar**[β]    **Tirth Vora**[β]
**Tushar Khot**[α]    **Ashish Sabharwal**[α]    **Peter Clark**[α]
[α]Allen Institute for AI    [β]OpenLocus    [γ]University of Massachusetts Amherst
**Website:** https://github.com/allenai/discoverybench
[*]equal contributions

## Abstract

Can the rapid advances in code generation, function calling, and data analysis using large language models (LLMs) help automate the search and verification of hypotheses purely from a set of provided datasets? To evaluate this question, we present DiscoveryBench, the first comprehensive benchmark that formalizes the multi-step process of data-driven discovery. The benchmark is designed to systematically assess current model capabilities in discovery tasks and provide a useful resource for improving them. Our benchmark contains 264 tasks collected across 6 diverse domains, such as sociology and engineering, by manually deriving discovery workflows from published papers to approximate the real-world challenges faced by researchers, where each task is defined by a dataset, its metadata, and a discovery goal in natural language. We additionally provide 903 synthetic tasks to conduct controlled evaluations on data-driven workflows that are not covered in the manually collected split. Furthermore, our structured formalism of data-driven discovery enables a facet-based evaluation that provides useful insights into different failure modes. We evaluate several popular LLM-based reasoning frameworks using both open and closed LLMs as baselines on DiscoveryBench and find that even the best system scores only 25%. Our benchmark, thus, illustrates the challenges in autonomous data-driven discovery and serves as a valuable resource for the community to make progress.

## 1 Introduction

Knowledge discovery via the scientific process has been a catalyst for human progress for centuries but has, thus far, been a predominantly manual pursuit (Glass & Hall, 2008). Recent breakthroughs in capabilities of large language models (LLMs) to reason and interface with the world using code (Chen et al., 2021; Roziere et al., 2023), external tools (Schick et al., 2024), and interactive agents (Yao et al., 2023; Majumder et al., 2023), however, now suggest the possibility of realizing a discovery system that is fully autonomous. Indeed, recent works (Majumder et al., 2024) provide initial evidence for this paradigm within the setting of *data-driven discovery*, where both search and verification of hypotheses may be carried out using a dataset alone (i.e., after physical experiments and data collection[1]), but the extent of this ability remains unclear. We, therefore, aim to systematically evaluate the following question:

> *How capable are current state-of-the-art LLMs at automated data-driven discovery?*

Answering this question is hard, as data-driven discovery in the wild (real-world) is diverse across domains and subject areas, which in turn makes it difficult to build a robust evaluation framework to measure progress. We address this using a pragmatic formalization of data-driven discovery, namely the search for a *relationship* that may hold between *variables* in a *context*, where (importantly) the description of those facets may not be in the language of the dataset. A data-driven discovery task

---

[1]In practice, experiments and analysis are interleaved, not sequential. Our concern in this work, however, is systematically studying the data analysis part of the (interleaved) pipeline.

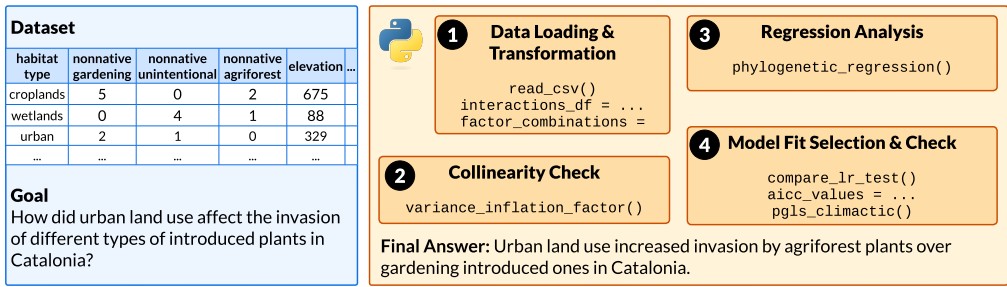

**DiscoveryBench Task**            **Ideal Discovery Agent Workflow**

Figure 1: Each DISCOVERYBENCH task consists of a goal and dataset(s) (left). Solving the task requires both statistical analysis and scientific semantic reasoning. The realistic requirement of domain-specific tools in some tasks adds more complexity. An ideal discovery system should implement discovery programs in Python.

then has one of these components missing, e.g., *"How did urban land use affect the invasion of introduced plants in Catalonia?"*. Importantly, this formalization allows for systematic, reproducible evaluation over a wide variety of real-world problems, by leveraging these facets.

Unlike prior datasets for statistical analysis (Liu et al., 2024) or AutoML (Zhang et al., 2023; Gijsbers et al., 2022), DISCOVERYBENCH tasks also require scientific semantic reasoning, for instance, deciding which of the many possible analysis techniques are appropriate for the domain (e.g., phylogenetic regression for plant invasion, Fig 1), how to clean and/or normalize the data, and how to map goal terms to dataset terms (e.g., "land use" to "habitat type"). Task solutions typically requires a multistep workflow (Fig 1, right). In this way, DISCOVERYBENCH is the first large-scale dataset to address the broader data-driven discovery pipeline, not just the statistical analysis component, and explore LLMs' capacity for this.

Given this framework, we created DISCOVERYBENCH by manually extracting 264 discovery tasks, i.e., goal + dataset(s), from over 20 published papers, as well as creating real-world discovery workflows that solve each task. We additionally provide 903 synthetic tasks across 48 domains generated using LLMs to mimic the real-world discovery process. The synthetic benchmark is complimentary, capturing a diverse range of data-driven workflows that could not be included in the manually collected split of our benchmark due to lack of reproducibility. Our contributions are thus:

- DISCOVERYBENCH, the first comprehensive benchmark to formalize the multi-step (code-based) process of data-driven hypothesis search and verification, covering many real-world discovery tasks plus additional synthetic tasks.
- A pragmatic formalism for data-driven discovery, flexible enough to characterize many real-world tasks while constrained enough to allow for rigorous, reproducible evaluation.
- A comprehensive evaluation across state-of-the-art LLM-based reasoning methods ("discovery agents"). We find performance peaks at 25%, demonstrating the challenging nature of our task.

These suggest that DISCOVERYBENCH may be a valuable resource for helping make progress on autonomous, data-driven discovery.

## 2 RELATED WORK

Automated data-driven discovery has been a long-standing dream of AI (Majumder et al., 2024; Kitano, 2016). Although there have been a range of **data-driven discovery systems**, from early ones that fit equations to idealized data, e.g., Bacon (Langley, 1981), to more modern ones handling complex real-world problems, e.g., AlphaFold (Jumper et al., 2021), their associated datasets are task-specific and customized to a pre-built pipeline. In contrast, DISCOVERYBENCH aims to be general, including testing whether systems can design appropriate workflows themselves over multiple tasks.

Several datasets and tools are available for **AutoML**, a related technology aimed at automating workflows for building optimal machine learning models (Jin et al., 2023; Zhang et al., 2023; LeDell & Poirier, 2020). AutoML tools include packages like Scikit (Feurer et al., 2015), and embedded in cloud platforms such as Google Cloud Platform, Microsoft Azure, and Amazon Web Services.

However, associated datasets for AutoML are primarily used for training models, rather than for open-ended discovery tasks.

Similarly, there are several datasets that test **statistical analysis** in various fields, e.g., (Shao et al., 2023; Li et al., 2024; Yang et al., 2022). Software packages like Tableaux, SAS, and R also support users in that task. However, these datasets and tools are designed specifically for data analysis, while DISCOVERYBENCH aims to automate the broader pipeline including ideation, semantic reasoning, and pipeline design, where statistical analysis is just one component.

| Feature | Ours | QRData | DSBench | BLADE |
|---|---|---|---|---|
| File handling | ✓ | ✓ | ✓ | ✓ |
| Statistical tools | ✓ | ✓ | ✓ | ✓ |
| Data Semantics | ✓ | ✗ | ✓ | ✓ |
| Hypothesis Discovery | ✓ | ✗ | ✗ | ✗ |
| Domain Knowledge | ✓ | ✗ | ✗ | ✓ |
| Domain-specific Tools | ✓ | ✗ | ✗ | ✗ |
| Scientific Hypothesis | ✓ | ✗ | ✗ | ✓ |

Table 1: Comparison with other data-analysis benchmarks

One recent dataset similar in spirit to ours is QRData (Liu et al., 2024). QRData also explores LLM capabilities but targets statistical/causal analysis for well-defined (mainly) textbook questions that have unique, (mainly) numeric gold answers. DSBench (Jing et al., 2024) proposes data science tasks that require agents to understand multimodal input files, generate complex code, and interpret and process them to solve a multiple-choice or short direct-answer task. Unlike DISCOVERYBENCH, these tasks do not require domain knowledge or hypothesis discovery. Recent work, BLADE (Gu et al., 2024), presents complex hypothesis verification as a multiple-choice QA task. Even though expected workflows are multi-step, at each step, the space of operations is limited to a small set, e.g., select, map, group-by, etc. In contrast, DISCOVERYBENCH has no prescribed boundaries on statistical techniques to apply, uses open-ended questions and answers, and complex tasks drawn from state-of-the-art published work. Further, DISCOVERYBENCH poses a high-level research question that requires an agent to do a hypothesis search in addition to verification. Table 1 summarizes a list of distinguishing features of DISCOVERYBENCH, which promotes a thorough, systematic evaluation of (LLM-driven) discovery systems in the challenging but realistic task of data-driven discovery.

## 3 FORMALIZATION

We begin by formalizing what we mean by a data-driven hypothesis and how the structure of a complex hypothesis may be viewed as a hypothesis semantic tree.

A **data-driven hypothesis** $h$ in $\mathcal{H}$ (the space of such hypotheses) is a declarative sentence about the state of the world whose truth value may be inferred from a given dataset $D$ using a verification procedure $\mathcal{V}_D : \mathcal{H} \rightarrow \{\text{supported}, \text{unsupported}\}$. For our benchmark, the space of valid $\mathcal{V}_D$ is potentially any executable Python program.

Inspired by recent work of Thompson & Skau (2023), we additionally introduce a structured formalism that breaks a hypothesis down into **three hypothesis dimensions**:

- **Contexts** ($c$): Boundary conditions that limit the scope of a hypothesis. E.g., *"for men over the age of 30"* or *"in Asia and Europe"* or unbounded/full dataset when not specified.
- **Variables** ($v$): Known set of concepts that interact in a meaningful way under a given context to produce the hypothesis. E.g., `gender`, `age`, or `income`. Note that each hypothesis is associated with a target variable and a set of independent variables.
- **Relationships** ($r$): Interactions between a given set of variables under a given context that produces the hypothesis. E.g., *"quadratic relationship"*, *"inversely proportional"*, or piecewise conditionals.

With slight abuse of notation, we can now equivalently define hypothesis $h := \psi(c, v, r)$, where $\psi(\cdot, \cdot, \cdot)$ is a generative function that returns the declarative sentence *"under context $c$, variables $v$ have relationship $r$."* For instance, for the hypothesis from Fig 1, $c := $ *"urban land use; plant type agriforest and gardening"*, $v := $ *"invasion rate"*, and $r := $*"invasion rate is higher in agriforest plants than gardening plants."* Note that the hypothesis itself is a sentence, not a tuple.

## 4 DISCOVERYBENCH

We now introduce a novel benchmark, DISCOVERYBENCH, for discovering data-driven hypotheses. In this benchmark, a *data-driven discovery task* is defined as follows: Given one or more task

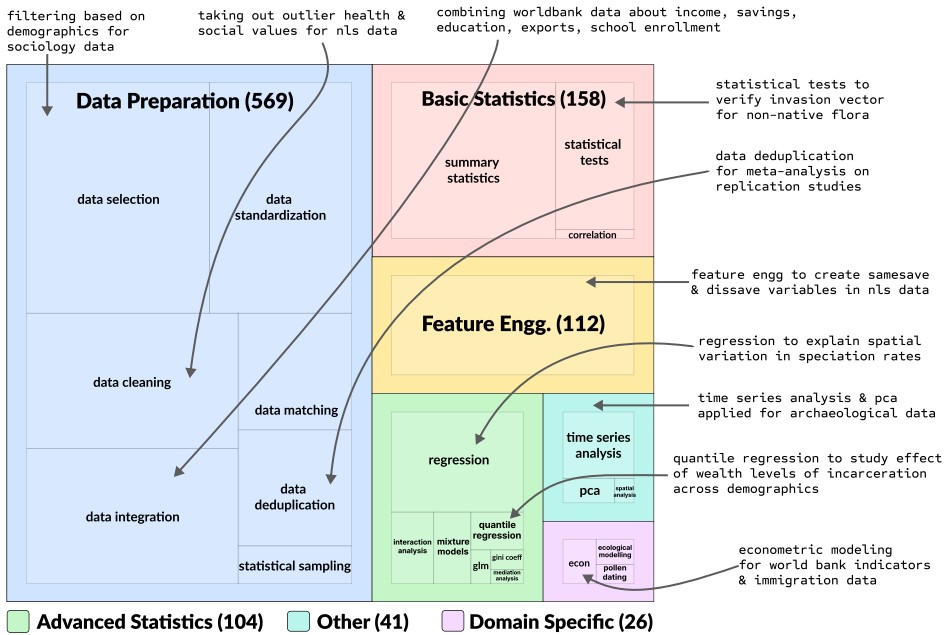

Figure 2: Workflow categories in DB-REAL with representative examples.

dataset(s) $D$ and a discovery goal $G$, derive a hypothesis $h$ addressing $G$ with the highest specificity for the context $c$, variables $v$, and relationship $r$ supported by $D$. Optionally, a workflow for deriving a hypothesis can be output to augment information already present in the hypothesis. DISCOVERYBENCH has two components: **DB-REAL** encompassing data-driven hypotheses and workflows derived from published scientific papers and **DB-SYNTH** capturing systematic variations in data-driven hypotheses and workflows obtained from synthetically generated datasets. The benchmark is released at: `https://github.com/allenai/discoverybench`. The licenses permitting the re-distribution of datasets collected from past work are provided in Appendix B and Table 3.

## 4.1 DB-REAL: COLLECTING DATA-DRIVEN HYPOTHESES IN THE WILD

Our goal is to replicate the scientific process undertaken by researchers to search for and validate a hypothesis from one or more datasets. We focus on six scientific domains where data-driven research is the cornerstone of scientific progress: sociology, biology, humanities, economics, engineering, and meta-science. Our gold trajectories to solve a discovery task carefully follow the published papers' workflows in respective domains. As most of the papers are highly cited, peer-reviewed, and from top venues from the domains, it is reasonable to assume the published workflows are scientifically valid.

To ensure that we correctly reproduced the methods and code implementations and reached the correct hypotheses: (a) We restrict ourselves to cases where the complete code was available (e.g., in GitHub) from the original authors, so we could directly reuse it; (b) For the Economics and Sociology related hypotheses, we consulted domain experts (Economics PhDs) to verify that our interpretation, implemented workflows, and replicated hypotheses make sense; (c) For ML engineering, we relied on our own expertise in the domain. See Appendix B for the original sources of data and workflows.

While validating novel discoveries is incredibly challenging and requires domain experts to verify, the goal of DISCOVERYBENCH is not to make new discoveries but rather to test if a discovery system can at least make known, validated discoveries correctly as a first step to the broader goal of automated discovery—an easier bar, and one which our benchmark can assess systematically. We detail our data collection process, which follows either a **data-first** or **code-first** approach.

For the **data-first approach**: 1) we filter papers based on open public datasets ($D$) such as National Longitudinal Surveys (NLS), Global Biodiversity Information Facility (GBIF), and World Bank Open Data (WBOD) that have workflow details; 2) we then try to replicate these workflows in Python. For this data-first approach, replication took up to 90 person-hours per dataset, often (30%) not resulting

in success. This highlights building data-driven discovery benchmarks from real studies is not only challenging and time-consuming, but automating discovery can also be key for scientific progress and reproducibility.

The data-first approach by design is limited to well-known aforementioned public datasets. To improve diversity in domains, datasets ($D$), and workflows, we also adopted a **code-first approach** to look beyond popular public datasets. In this approach, we 1) search for code repositories based on scientific papers with available datasets and 2) attempt to replicate them in Python with existing code or from scratch with interpretation of the associated paper. We looked at 785 data points in Zenodo, EU's Open Research Repository, with a filter for computational notebooks. Over 85% of the repositories either had missing code, code that could not be easily translated to Python, or a proprietary/non-open dataset. We finalized a candidate list of 14 repositories, but in the end, 3 of them passed all our checks for their hypotheses to be included in the benchmark[2].

Upon replication of the result or implementation of the full procedure as described in the paper, we include the (dataset $D$, hypothesis $h$, implementation workflow[3]) tuple to the benchmark.

During the process, the implementation workflow may lead to other hypotheses that are not directly reported in the paper but can be supported by the data. We included them in DISCOVERYBENCH, which leads to a good mix of already reported science-worthy hypotheses as well as novel hypotheses grounded in datasets. This is particularly useful as our goal is to evaluate LLMs' ability to solve a discovery task that is realistic but never reported before.

Finally, the task datasets are supplemented with a dataset description, natural language descriptions of the columns, and additional background knowledge related to the domain or the datasets. Some of our tasks, for instance, archaeology, require domain knowledge to derive a particular hypothesis.

**Inferring task difficulty.** Inferring the hypothesis space given a task dataset alone is impractical due to incomplete *a priori* information about unobserved intermediate variables and their association with observed ones. To infer task difficulty, we, therefore, use the length of the implementation workflow required to derive a target hypothesis. Each step in the workflow contributes to 1 unit length to workflow length. In some cases, we derive two tasks: easy and hard from the same hypothesis, where for easy, we provide the derived variables as observed variables in the dataset (e.g., BMI), and for hard, it would require deriving intermediate variables (BMI from height and weight) to reach the target. Intuitively, discovery becomes harder as the hypothesis search space increases. In practice, this setting is observed when a task requires access to multiple datasets.

**Forming discovery goals.** By definition, each hypothesis can be fully specified by the declarative sentence as $h := \psi(c, v, r)$. To systematically construct the discovery goals for the task, we first mask one of each dimension, context $c$, variable $v$, relationship $r$, and generate a discovery goal to identify the masked information given the rest of the hypothesis and the task dataset(s). For instance, for a target hypothesis, *"The effect of socioeconomic status on college degree completion is higher for females (0.4995) than males (0.4467)"*, we form a discovery goal as *"How does socioeconomic status impact on college degree completion for females compared to males?"* seeking the relationship $r$ to be discovered from the dataset(s) given the relevant variables $v$ and context $c$. Additionally, we ensure each discovery goal leads to only one answer, i.e., the target hypothesis.

### 4.1.1 FEATURES OF DB-REAL BENCHMARK

DISCOVERYBENCH incorporates a broad landscape of data-driven discovery. With over 500 instances of data preparation activities such as cleaning, deduplication, and integration, captures the complexity of real-world data preprocessing for discovery. Tasks also demand a spectrum of statistical methods, from statistical tests to mixture models, and include domain-specific approaches in econometric and ecological modeling, as reflected in the Fig 2[4].

|  | Train | Test |
|---|---|---|
| # tasks | 25 | 239 |
| # unique hypotheses | 14 | 144 |
| # tasks need $> 1$ dataset | 4 | 110 |
| # domains | 3 | 6 |

Table 2: Statistics for DB-REAL

---

[2]Some repositories include hypotheses from multiple papers as their background.

[3]Implementation workflows are descriptions of implementations to reproduce the results from original works.

[4]A task may require multiple data preparation and analytical activities.

Table 2 shows the diversity of tasks both in train and test split for DB-REAL. Most importantly, the benchmark incorporates 114 (4 + 110) tasks that require more than one related datasets to be analyzed, with a maximum of 6 datasets for a task. Each workflow within the dataset can be viewed as a composition of unit actions—such as code generation for statistical tests—that LLMs excel at, showing how our tasks require the chaining of such atomic actions to address complex scenarios for data-driven discovery. We measure the complexity of these workflows by quantifying the number of unit actions involved, referring to this as the *workflow length*, whose distribution can be seen in Fig 5.

## 4.2 DB-SYNTH: GENERATING DATA-DRIVEN HYPOTHESES USING LLMS

While DB-REAL covers a broad set of domains and workflows, several other types of workflows (e.g., nonlinear regression, cluster analysis) could not be included in the benchmark due to difficulty in replication or out-of-scope domains. However, to provide a comprehensive view of data-driven discovery, we introduce a *complimentary* split of our benchmark, DB-SYNTH, which captures synthetic task examples constructed from inspirational workflows from published works (Galbraith et al., 2010). Note, the workflow categories (e.g., symbolic regression, equation discovery, and function approximation) in DB-SYNTH are taken from a broader set of papers beyond what we considered in DB-REAL; hence the difficulty and solutions of the tasks broadly vary from the real split of our benchmark.

Our goal is to reverse-engineer the process of hypothesis discovery to synthesize datasets and discovery tasks that require analysis workflows similar to those that are broadly observed in published works. Our approach leverages the broad pre-trained knowledge of LLMs in four stages:

**Domain sampling:** First, we prompt the LLM to generate a list of diverse *domains* along with their natural language descriptions. E.g., "Ancient architecture" → "Related to architectural marvels, ancient building and construction techniques." Note, such a domain is not covered in DB-REAL.

**Semantic tree construction:** For each domain, we then build a hypothesis semantic tree $\mathcal{T}_h$, which is a tree of variables depicting the dependencies between the variables in the dataset (leaf nodes), intermediate variables, and the target (dependent) variable of a primary hypothesis $h$ (root node). Note, $\mathcal{T}_h$ is not the hypothesis itself.

Specifically, we prompt the LLM with the domain and a sampled real-world workflow (e.g., "within-cluster analysis") to generate a hypothesis and its target variable. To encourage diversity in plausible (as per the LLM) workflows and to increase coverage (by accounting for uncovered workflows in DB-REAL), we allow LLMs to combine several real-workflow steps (type of variables, type of relationships, etc). This significantly improves the likelihood of generating synthetic workflows that closely

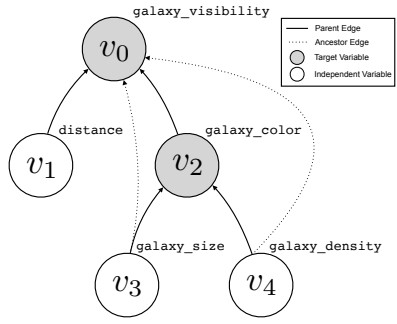

Figure 3: Hypothesis Semantic Tree

resemble the real ones. Setting the target variable as root (e.g., `galaxy_visibility`), we then derive child nodes by generating the independent variables required to verify $h$ using $\mathcal{V}(\cdot)$, as seen in Fig 3. We operationalize this by generating a column name and description for each child node (along with a data type and range) and a pandas expression[5] (Wes McKinney, 2010) over only independent variables in $\mathcal{T}_h$ such that its execution results in the target variable. We repeat this with each leaf in $\mathcal{T}_h$ as the root of a new semantic sub-tree, generating intermediate hypotheses and a new set of variables until the desired height of $\mathcal{T}$ is reached.[6] We also generate a set of distractor columns disjoint from nodes in $\mathcal{T}$, thus resulting in a synthetic semantic forest $\mathcal{F}$.

**Data generation:** We then construct a task dataset by generating synthetic data in a bottom-up manner (i.e., from leaves to root) for each node in $\mathcal{F}$. Starting with various sampling strategies for leaf nodes (see more in Appendix C), for each subsequent level in $\mathcal{F}$, we create new columns for nodes by simply executing their pandas expressions. Finally, to mimic real-world challenges in data

---

[5]The pandas expression encodes the structured hypothesis $\psi(c, v, r)$.

[6]In practice, with probability 0.6, we choose whether a node is derived further or marked as a leaf.

collection, we probabilistically perturb each instance $x \in \mathbf{x}_i$ by adding noise or dropping values to create missing data[7]. Note that at this stage, $D$ contains a column for each node in $\mathcal{F}$.

**Task generation:** For each internal node $h$ in $\mathcal{F}$, we now create multiple task datasets $D_h^{(l)}$ from $D$, varying the difficulty of the discovery task based on the path length $l$ between $h$ and the observed independent variables in $\mathcal{F}$. Finally, we follow the same strategy for goal formulation as DB-REAL. We generate 903 tasks over 48 diverse domains and assign them to development and test sets using an 80/20 split, where each task is additionally tagged with a difficulty level from 1-4.

See Appendix D for the compositional details on both real and synthetic splits of DISCOVERYBENCH. We also list some limitations and ethical considerations around our benchmark in Appendix A.

## 4.3  EVALUATION

We evaluate task performance by measuring the alignment of the predicted and gold hypotheses in natural language. We take inspiration from recent works in LLM benchmarking (Shashidhar et al., 2023; Zeng et al., 2023; Yuan et al., 2023; Fu et al., 2023; Li et al., 2023; Lin et al., 2024) and design a model-based evaluation strategy using `gpt-4-preview-0125` as the *evaluator*, conditioned on our structured formalism of data-driven hypotheses. Critically, the evaluator assesses entailments/equivalences between linguistic elements of a predicted and gold hypothesis pair, folowing from several LM-based language entailment as automatic tools for scientific claim verification (Wadden et al., 2022; Min et al., 2023; Schlichtkrull et al.).

Recall the declarative form $h := \psi(c, v, r)$ of a hypothesis $h$ that constitutes of three hypothesis dimensions: context ($c$), variables ($v$), and relationships ($r$). Given this structured representation of the gold and predicted hypotheses, we then compute a **hypothesis match score (HMS)**, which measures the degree to which two hypotheses align on each dimension, as follows:

- **context:** We first use our GPT-4 based evaluator to independently extract the contexts for the gold ($h^g$) and predicted ($h^p$) hypotheses (prompt in Listing 1). Then, we compute a context alignment score $\mathrm{ctx_{align}}$, which takes value 1 if a GPT-4 based evaluator deems the contexts extracted from $h^g$ and $h^p$ as semantically equivalent, otherwise 0 (prompt in Listing 2). Note, instead of comparing the full hypotheses, we employ a much simpler faceted textual entailment (or semantic similarity) problem between two text segments, which GPT-4 is very strong at (Sanyal et al., 2024).
- **variables:** Using our LLM evaluator, we extract the set of interacting variables in the gold and predicted hypotheses using the GPT-4 based evaluator (prompt in Listing 3). We compute the alignment between these two sets of variables as an F1 score, $\mathrm{var_{F1}}$, capturing how aligned the variables from $h^g$ with the variables from $h^p$.
- **relationships:** Our GPT-4 evaluator computes a relationship accuracy with reference to the relationship between the gold variables ($\mathrm{rel_{acc}}$) based on evaluator judgments using the following scoring heuristic: 1 if there is an exact match of the relation, 0.5 when the predicted relationship is broader than the gold relationship but encompasses it , and 0 otherwise (prompt in Listing 4).

Two authors independently evaluated the step-by-step LLM-based evaluation process on 200 examples. After resolving disagreements, we find GPT-4 was 97% correct in context extraction, 99% correct in context matching, 94% correct in variable extraction, and 99% correct in assessing relationship accuracy, exhibiting sufficiently strong performance in entailment-based evaluation. Finally, we compute HMS (scaled) $\in [0, 100]$ as the alignment of the variable and relationship dimensions over context-matched hypotheses weighted by the overall context alignment:

$$\mathrm{HMS}(h^p, h^g) = \mathrm{ctx_{align}}(h^p, h^g) \times \mathrm{var_{F1}}(h^p, h^g) \times \mathrm{rel_{acc}}(h^p, h^g)$$

**Is the outcome-based evaluation sufficient?** Yes. The gold hypotheses, which a discovery agent does not see at the test time, are intentionally designed to be very specific, making it highly improbable to arrive at them by chance or via invalid means. E.g., while a discovery agent might guess a positive correlation via invalid means, it would be unable to compute the precise target *correlation coefficient* without using the correct statistical analysis. The HMS metric is designed to ensure that all aspects of the target discovery are satisfied in order to receive full reward. Most significantly, the very low evaluation scores (in Fig 4) substantiate this—even the best agents perform poorly, suggesting that random guess or invalid shortcuts do not lead to the target result.

---

[7]Each value is noised independently; therefore, each row has sufficient true data useful for discovery.

|  | GPT-4o | GPT-4p | Llama-3 |
|---|---|---|---|
| **DB-REAL** | | | |
| NoDataGuess | 0.0 | 4.7 | 11.5 |
| CodeGen | 20.9 | 19.3 | 18.9 |
| React | 24.4 | 22.4 | 18.4 |
| DataVoyager | 20.8 | 20.4 | 11.6 |
| Reflexion (Oracle) | **24.6** | **23.1** | **22.5** |
| **DB-SYNTH** | | | |
| CodeGen | 14.1 | 8.7 | 10.9 |
| React | 11.6 | 7.4 | 12.0 |
| DataVoyager | 10.1 | 6.9 | 11.7 |
| Reflexion (Oracle) | **15.7** | **12.9** | **23.2** |

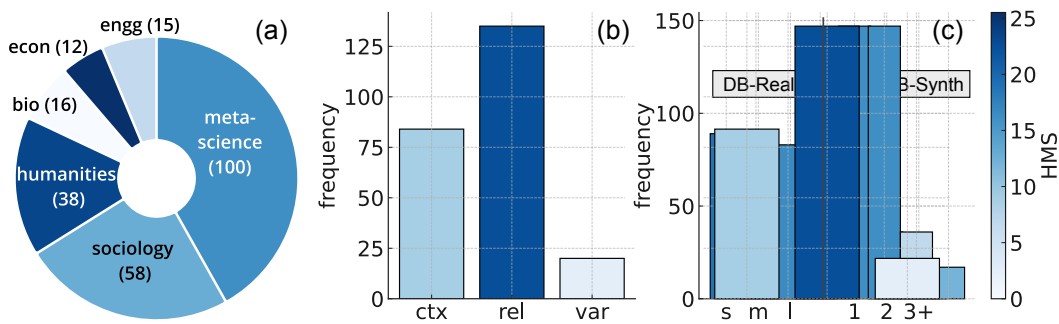

Figure 4: (Left) Hypothesis Matching Scores (HMS) across agent-LLM pairs in DB-REAL and DB-SYNTH. (Right) Scatter plots for mean $ctx_{align}$ vs mean $var_{F1}$ and mean $var_{F1}$ vs mean $rel_{acc}$. The plots show that agent-LLM combinations that are capable of identifying the correct context have a higher likelihood of obtaining the right set of interacting variables. Similarly, identifying correct variables leads to better identification of relationships between them.

Figure 5: Best non-oracle agent's performance (HMS) (a) across domains, (b) for goal types (dimension to be discovered), and (c) for different workflow lengths. In (c) workflow length categories for DB-REAL are s: $< 10$, m: $> 10, < 20$, l: $> 20$. For DB-SYNTH, it is the semantic tree height.

**Human alignment with HMS.** To assess the robustness of the HMS metric, we ask 3 annotators to provide a preference ranking over 2 predicted hypotheses for 100 such random pairs (from DB-REALtest set) when compared to their respective gold hypotheses of each task. We then compute the corresponding preference ranking using our HMS metric and compare the two lists of preference rankings. We find human annotators agree with the HMS-based ranking 95% of the time. Each pair is evaluated by three annotators (and a majority alignment verdict is taken), with a Fleiss kappa of 0.91, showing a very good strength of agreement among annotators. For the null hypothesis, "there is a difference between human and HMS rankings," the power of the test with 100 sample pairs is $< 0.05$. Upon testing, we reject the null hypothesis, i.e., we find the rankings obtained by HMS are statistically the *same* ($p < 0.001$) as the rankings provided by human annotators. Our finding here supports the evidence in literature on human alignment in LM-based evaluation for scientific claim verification (Wadden et al., 2022; Schlichtkrull et al.).

## 5 EXPERIMENTS

### 5.1 DISCOVERY AGENTS

We benchmark state-of-the-art LLM-based few-shot reasoning methods as discovery agents with two closed models, GPT-4o and GPT-4-0125-preview (GPT-4p), and one open, Llama-3-70B, model powering the reasoning methods. A discovery agent takes the task description, paths to the task dataset(s) $D$, metadata about the datasets (description, column descriptions), and the goal, $G$, to produce a natural language (NL) hypothesis specified by context, variables, and relationship.

- **CodeGen** generates the entire code at one go to solve the task, where we provide a demonstration of a solution code in the context. After code execution and based on the result, it generates the NL hypothesis and summarizes the workflow.

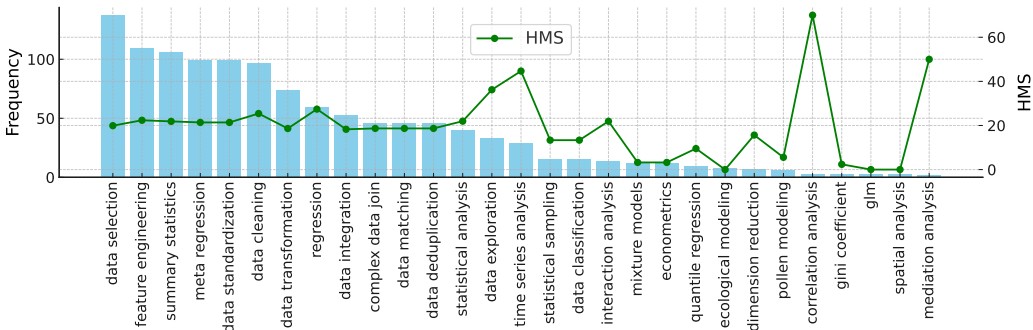

Figure 6: Breakdown of performance (HMS) of the best non-oracle agent in several workflow categories in their decreasing order of appearance in the benchmark.

- **ReAct** (Yao et al., 2023) solves the task by generating thought and subsequent codes in a multi-turn fashion based on the past thought-action-observation trace.
- **DataVoyager** is a multi-component data-driven discovery agent from (Majumder et al., 2024). It has four components, planner, code generator, data analyzer, and critic, that orchestrate the discovery process.
- **Reflexion (Oracle)** (Shinn et al., 2023) is an extension of CodeGen agent, where at the end of one trial, we provide the "oracle" HMS score as an evaluation signal, and it generates a reflection to improve (when HMS $< 1$) in the next trial till it solves the task, or maximum trials (3) are reached.
- **NoDataGuess** guesses the hypothesis (in DB-REAL) just from the dataset description and the goal without accessing the datasets where we measure LLM's memorization of already published works.

## 5.2 MAIN RESULTS

Fig 4(left) shows that overall performance for all framework-LLM pairs is low for both DB-REAL and DB-SYNTH, highlighting the challenging nature of the task and the benchmark. Most importantly, effective reasoning prompts such as React and planning with a self-critic (DataVoyager) do not help improve the simple CodeGen agent. But with oracle feedback, Reflexion (Oracle) significantly improves over CodeGen (base) performance. Analysis reveals that almost all non-reflexion agents solve the *easiest* (in terms of workflow category and length) instances from the benchmark. GPT-4o refuses to hallucinate in the NoDataGuess baseline, whereas surprisingly, Llama-3 performs similarly in both data and no-data modes. We additionally observe that agents that perform well on DB-REAL may not perform equally (well) in DB-SYNTH, possibly due to the differentiating nature of underlying workflows. For e.g., while it is easier to perform linear regression for discovery agents, performing non-linear regression embedded in DB-SYNTH is hard.

## 5.3 DISCUSSION

**Context is important.** Fig 4(right) shows score trends in the three-pronged discovery process. A positive trend between $ctx_{align}$ and $var_{F1}$ signifies that to predict variables accurately, accurate context prediction is necessary. However, correct identification of context is an important first step, although it does not guarantee success. Similarly, predicting the variables correctly increases the chance of predicting relationships among them. Additionally, we find clusters (e.g., Llama-3 has consistently lower scores) in terms of dimension-level performance around LLMs-agent frameworks. Also, across LLMs, a very similar trend is observed as they operate from different agent frameworks.

**Performance across domains and goal types.** Fig 5(a) depicts that biology (0%) and engineering (7%) perform the worst due to their higher dependence on advanced statistical methods, while economics (25%) and sociology (23%) perform better. Additionally, Fig 5(b) shows goals related to discovering a relationship given context and variables are more easily solved than the other two types of goals, finding context and variables. This is explained by the complexity of the hypothesis search, which is broader for finding the right context or a set of variables given a fixed relationship, whereas finding the relationship given context and variables is easier.

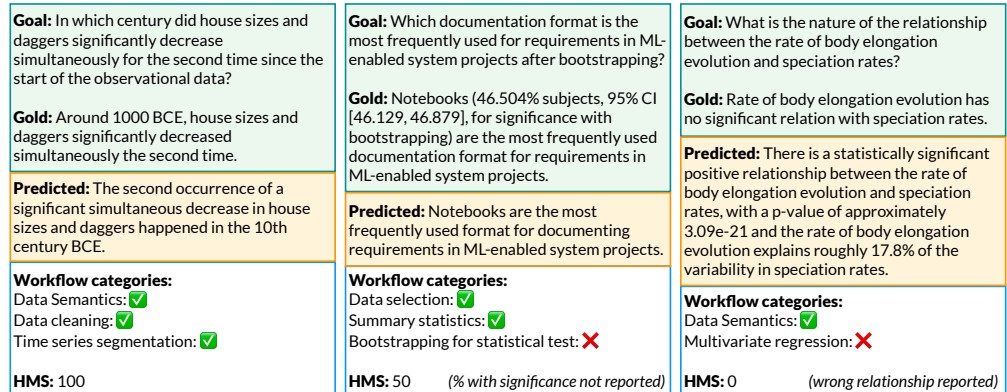

**Goal:** In which century did house sizes and daggers significantly decrease simultaneously for the second time since the start of the observational data?

**Gold:** Around 1000 BCE, house sizes and daggers significantly decreased simultaneously the second time.

**Predicted:** The second occurrence of a significant simultaneous decrease in house sizes and daggers happened in the 10th century BCE.

**Workflow categories:**
Data Semantics: ✅
Data cleaning: ✅
Time series segmentation: ✅

**HMS:** 100

---

**Goal:** Which documentation format is the most frequently used for requirements in ML-enabled system projects after bootstrapping?

**Gold:** Notebooks (46.504% subjects, 95% CI [46.129, 46.879], for significance with bootstrapping) are the most frequently used documentation format for requirements in ML-enabled system projects.

**Predicted:** Notebooks are the most frequently used format for documenting requirements in ML-enabled system projects.

**Workflow categories:**
Data selection: ✅
Summary statistics: ✅
Bootstrapping for statistical test: ❌

**HMS:** 50        *(% with significance not reported)*

---

**Goal:** What is the nature of the relationship between the rate of body elongation evolution and speciation rates?

**Gold:** Rate of body elongation evolution has no significant relation with speciation rates.

**Predicted:** There is a statistically significant positive relationship between the rate of body elongation evolution and speciation rates, with a p-value of approximately 3.09e-21 and the rate of body elongation evolution explains roughly 17.8% of the variability in speciation rates.

**Workflow categories:**
Data Semantics: ✅
Multivariate regression: ❌

**HMS:** 0        *(wrong relationship reported)*

Figure 7: Representative examples of tasks where we qualitatively evaluate non-oracle best agent's trajectories along with their performance indicated by HMS.

**Domain knowledge dependency.** To check if additional domain knowledge helps agents perform better, we collect targeted domain knowledge for the archaeology-related tasks that needed significant domain knowledge during data collection. When added as additional hints, we find that DataVoyager's (GPT-4p) performance jumps from 9.9% (w/o domain knowledge) to 25.4% (w domain knowledge) for archaeology tasks.

**Workflow complexity barrier.** Almost all agents struggle more with tasks involving complex statistical techniques, complex data preparation methods, or domain-specific models. The top three workflow categories where the best non-oracle model was highly performant are correlation analysis, time series analysis, and social-science-related analysis, whereas the lowest three workflow categories are spatial analysis, pollen modeling, and ecological modeling. Fig. 6 shows overlayed HMS on several workflow categories with several informative spikes and drops. Simpler, common workflow categories (e.g., correlation, regression, data cleaning) see performance spikes, whereas long-tail workflow categories pose a great challenge even to the best discovery agent. The richness and the requirement of long-tail analysis distinguish our benchmark from the traditional data-analysis benchmarks, providing a systematic way to evaluate LLMs' capability in realistic data-driven discovery.

**Impact of workflow length.** Inherently, the difficulty of the tasks is measured by the gold workflow length (DB-REAL) or the height of the semantic tree (DB-SYNTH). Figure 5(c) shows a decreasing trend in performance as workflow length (hence, complexity) increases. The performance drops significantly even for medium-length workflows, highlighting current agents' limitations.

**Qualitative (error) analysis.** Fig. 7 depicts three examples where the best non-oracle agent shows varied performance. While the common tendency of LLMs is to arrive at *an* answer to a goal as soon as possible, this behavior appears to be costly for our benchmark. For example, when asked to perform additional significance tests to verify the importance of summary statistics, the agent does not pursue it consistently, resulting in low scores. Very often, most discovery agents will not perform thorough reasoning while solving a task from the benchmark. For example, when several predictive factors coexist, the relation between two variables should be validated by taking all of them into account (e.g., via a multivariate regression). However, the impatient discovery agents register simpler analysis, which (in Fig. 7 example) yield either underspecified or wrong answers.

## 6    CONCLUSION

We present DISCOVERYBENCH, the first data-driven discovery benchmark consisting of 264 discovery tasks that capture real scientific workflows extracted from published works. We supplement this with 903 structurally generated synthetic tasks, tailored to evaluate discovery agents for complimentary workflows. We benchmark state-of-the-art reasoning frameworks with the most advanced LLMs, but the best agent's performance only peaks at 25% underscoring the challenging nature of the task and the benchmark. We hope our timely contribution can increase interest and efforts in making progress on reliable and reproducible autonomous scientific discovery using large generative models.

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

## A  APPENDIX

## A  LIMITATIONS

We currently filtered domains and tasks that required forecasting, simulation, or very specific modeling (species distribution, infection spread, astrophysics equations for exoplanets) in the DB-REAL benchmark as they were very time-consuming to replicate as well as discover hypotheses. As a result, we discarded more papers focused on natural and physical sciences compared to social sciences, which we plan to include in future benchmarks.

We currently do not tackle the challenge of understanding and processing massive datasets, such as the 8.92 petabytes data from the Cancer Genome Atlas (`https://portal.gdc.cancer.gov`). While the potential to discover new insights from such vast data volumes is significant, ensuring these findings are robust and not subject to $p$-hacking remains unaddressed by our current methods.

We currently do not handle multi-modal data and complex pipelines, such as those needed for analyzing satellite and other geospatial data relevant to climate science and astronomy data. This would involve multiple stages of data processing, the use of various tools, and managing workflow complexities, for example, analyzing thousands of species patterns combined with satellite data to study habitats.

**Ethical Considerations**   There could be many potential societal consequences of systems tuned on our proposed benchmark since it involves using LLMs, such as policy misuse, legal ramifications, and false discovery. On the positive side, our proposed benchmark can advance the rate of discovery, leading to an improved standard of living and social well-being.

## B  DATA COLLECTION FOR DB-REAL

For data-first approach, replication took 15 to 40 person-hours for each NLS-related paper and up to 90 person-hours for the GBIF dataset, where specialized domain knowledge and tools led to higher complexity. All papers replicated in the NLS dataset were included, while less than half of the papers in specialized datasets like GBIF and WBOD were added to DISCOVERYBENCH.

**Citation/Repositories for DB-REAL:** List of scientific works from where we have replicated our gold workflows and hypotheses:

1. Sociology: (Zaw et al., 2016; Apel & Sweeten, 2010; Alexander et al., 1982; Smith et al., 2005; Dougherty, 2003)

2. Biology: (Cerezer et al., 2023; Riera et al., 2024)

3. Economics: (Pal, 2023; Appiah, 2017; Weatherly et al., 2022; Rambeli et al., 2021; Ottaviano et al., 2013)

4. Engineering: (Alves et al., 2023)

5. Meta-science: (Heyard & Held, 2024)

6. Humanities: (Brozio et al., 2024; 2019; Lorenz, 2018; Maida et al., 2023; Sommerfeld, 2013; Feeser et al., 2019; Parkinson et al., 2021; Kneisel, 2021; Kaiser & Schier, 2021; Brinkmann, 2019; Dal Corso et al., 2019; Palmisano et al., 2021; Parkinson et al., 2021)

**Distribution:**  All assets come under CC license or open licenses (Table 3). All NLS-related datasets are licensed under CCZero, and the rest of the datasets are licensed under CC by 4.0, which allows them to be freely used/distributed as long as they are properly cited. In NLS and ML engineering datasets, where there are human subjects, datasets are anonymized to protect respondent confidentiality.

## C  DATA GENERATION FOR DB-SYNTH

For leaves, we use different sampling strategies based on the data type. Specifically, for categorical nodes, we sample instances with replacement from the range of allowed values, whereas for numeric,

| Dataset | Paper/Citation | License | Source |
|---|---|---|---|
| `nls_raw`, `nls_bmi_raw`, `nls_bmi`, `nls_ses`, `nls_incarceration` | National Longitudinal Surveys (Moore et al., 2000) | Creative Commons CCZero | Link |
| `introduction_pathways_non-native_plants` | Effect of introduction pathways on the invasion success of non-native plants along environmental gradients (Riera et al., 2024) | CC by 4.0 | Link |
| `worldbank_education_gdp`, `worldbank_education_gdp_indicators` | WorldBank Data `https://databank.worldbank.org/source/world-development-indicators` | CC by 4.0 | Link |
| `archaeology` | Patterns of Socio-economic and Cultural Transformations in Neolithic and Bronze Age Societies in the Central Northern European Plain (Brozio et al., 2024) | CC by 4.0 | Link |
| `meta_regression_raw`, `meta_regression` | Meta-regression to explain shrinkage and heterogeneity in large-scale replication projects (Heyard & Held, 2024) | CC by 4.0 | Link |
| `immigration_offshoring_effect_on_employment` | Immigration, Offshoring, and American Jobs (Ottaviano et al., 2013) | CC by 4.0 | Link |
| `evolution_freshwater_fish` | Accelerated body size evolution in upland environments is correlated with recent speciation in South American freshwater fishes (Cerezer et al., 2023) | CC by 4.0 | Link |
| `requirements_engineering_for_ML_enabled_systems` | Industrial Practices of Requirements Engineering for ML-Enabled Systems in Brazil (Alves et al., 2024) | CC by 4.0 | Link |

Table 3: Licenses for original data sources. The Creative Commons Attribution license allows the re-distribution and re-use of a licensed work on the condition that the creator is appropriately credited.

we first select a distribution (e.g., normal) and its parameters based on the specified range and then perform sampling. For each subsequent level in $\mathcal{F}$, we create new columns for nodes by simply executing their pandas expressions[8]. To recover from any execution errors, we additionally use a self-refine (Madaan et al., 2023) approach to generate new pandas expressions guided by the execution error logs. Finally, to mimic real-world challenges in data collection, we probabilistically perturb each instance $x \in \mathbf{x}_i$ by adding noise or dropping values to create missing data[9]. After generation, $D$ contains a column for each node in $\mathcal{F}$.

The expressions associated with the hypothesis themselves are generated by prompting the model with natural-language workflow steps, e.g., "within-cluster analysis" and "temporal analysis." We first curate a pool of real workflow steps from DB-Real workflows. Then to encourage diversity in plausible (according to the LLM) workflows and to increase coverage (by accounting for workflows that we could not cover in DB-Real for practical/reproducibility reasons), we allow LLMs to combine several real-workflow steps (type or variable derivation, type of relationship derivation, etc) plausibly through drawing from the real workflow pool. This significantly improves the likelihood of testing

---

[8]The expression is guaranteed to only have variables already generated due to the bottom-up construction.
[9]Each value is noised independently; therefore, each row has sufficient true data useful for discovery.

synthetic workflows that connect well with the real ones. Moreover, we also use self-refine and human filtering to monitor the quality of the expressions and the variable metadata therein after tree and data generation. E.g., we saw a substantial improvement in quality when generating variable values from a specified range rather than a random range.

## D    COMPOSITION OF DISCOVERYBENCH

### D.1    METADATA STRUCTURE

- **id**: An identifier for the metadata.

- **domain**: The broad field of study or area of research.

- **workflow_tags**: A set of keywords summarizing the main processes or techniques used in the replication implementation. They provide an overview of the methodological approach and facilitating the identification of relevant analytical techniques.

- **domain_knowledge**:
    - Contextual information or insights related to the dataset, explaining how certain behaviors or variables can be interpreted within the field of study.
    - It helps open avenues to think in directions that LLM might not have considered otherwise, broadening the understanding of the field.

- **datasets**: Contains detailed information about the datasets used, including:
    - **name**: The name or filename of the dataset.
    - **description**: A summary of the dataset's contents and the type of data it includes.
    - **max_depth**: The maximum hierarchical level of nested data structures within the dataset, indicating the complexity of the data.
    - **columns**: Detailed descriptions of each column in the dataset, including:
        * **name**: The column's name or header.
        * **description**: Explanation of the data contained in the column and its significance.
        * **depth**: The hierarchical level of the column within the dataset, indicating its structural position.

- **hypotheses**: Statements or predictions being tested, divided into:
    - **main**: Primary hypotheses that are central to the discovery task.

- **workflow**: A step-by-step description of the replication process followed to validate the hypotheses, outlining the methods and procedures used from data preparation to final analysis. Some of the workflows and sub-workflows are high-level and thus the same for different queries as they follow the same implementation leading to a range of hypotheses.

- **queries**: Goals related to each hypothesis, each including:
    - **qid**: A unique identifier for the goal for a given true/gold hypothesis.
    - **difficulty**: Categorization of the difficulty. Structurally defined for DB-SYNTH using the semantic tree definition.
    - **true_hypothesis**: The hypothesis being tested through the goal. This defines the primary statement or prediction under investigation.
    - **relevant_cols**: Columns from the dataset that are relevant to answering the query, indicating the specific data points that can be used in the analysis. Only appears for DB-SYNTH.
    - **target_col**: The column being predicted or the dependent variable in the analysis. Only appears for DB-SYNTH.
    - **question_type**: The type of question being asked categorizing the nature of the inquiry.
    - **question**: The discovery goal.

## D.2 DIRECTORY STRUCTURE FOR DB-REAL

There may be more than one query per metadata. The train split contains 14 metadata files and 25 queries. The test split contains 144 metadata files and 239 queries. Metadata folders with the same prefixes use the same underlying dataset with either a subset or a preprocessed version. When dealing with a full dataset (i.e., nls_raw), the task becomes substantially harder due to the data preparation required.

```
|-test
|---archaeology
|---introduction_pathways_non-native_plants
|---meta_regression
|---meta_regression_raw
|---nls_incarceration
|---nls_raw
|---nls_ses
|---requirements_engineering_for_ML_enabled_systems
|---worldbank_education_gdp
|---worldbank_education_gdp_indicators
|-train
|---evolution_freshwater_fish
|---immigration_offshoring_effect_on_employment
|---nls_bmi
|---nls_bmi_raw
```

## D.3 DIRECTORY STRUCTURE FOR DB-SYNTH

There is one query per metadata. The train split contains 551 metadata files (queries), the dev split contains 153 metadata files (queries), and the test split contains 200 metadata files (queries).

```
|-test
|---ancient-languages_*_*
|---artificial-ecosystems_*_*
|---astronomy_*_*
|---board-games_*_*
|---coding-competitions_*_*
|---digital-artistry_*_*
|---futuristic-technology_*_*
|---impressionist-art_*_*
|---machine-learning_*_*
|---molecular-gastronomy_*_*
|---neuroscience_*_*
|---philosophical-debates_*_*
|---robotics_*_*
|-train
|---adventure-travel_*_*
|---ancient-architecture_*_*
|---ancient-astronomy_*_*
|---aviation_*_*
|---biodiversity-conservation_*_*
|---cryptic-puzzles_*_*
|---cryptocurrency_*_*
|---culinary-arts_*_*
|---cybersecurity_*_*
|---environmental-activism_*_*
|---fashion-design_*_*
|---fine-arts_*_*
|---literary-classics_*_*
|---marine-biology_*_*
```

```
|---marine-conservation_*_*
|---medieval-literature_*_*
|---musical-therapy_*_*
|---photography_*_*
|---robotic-explorers_*_*
|---solar-power_*_*
|---space-tourism_*_*
|---steampunk-culture_*_*
|---theater-productions_*_*
|---underwater-archaeology_*_*
|---urban-gardening_*_*
|---vintage-automobiles_*_*
|---virtual-reality_*_*
```

# E  EXPERIMENTS

For GPT-based models, we use OpenAI API (`https://platform.openai.com/docs/models`), and for Llama3, we used Together API (`https://docs.together.ai/docs/inference-models`)

## E.1  DISCOVERY AGENT

The command `discovery_agent.py` is used with various options to customize its behavior for discovery tasks. Below are the options explained:

- **Usage:** `discovery_agent.py [OPTIONS] QUERY` – Executes the discovery agent with specified options.
- **Options:**
  - `-agent_type [coder|react]`: Specifies the type of agent to use for discovery. The default type is `coder`. Options include `coder` for code-related tasks and `react` for reactive tasks.
  - `-model_name TEXT`: Sets the model to be used. The default is `gpt-4o`. Available models include `gpt-4-turbo`, `llama-3-70b-chat`, `claude-3-opus`, and `gemini-pro`. An exhaustive list is available in `config/model_config.json`.
  - `-api_config TEXT`: Path to the API configuration file. The default path is `config/api_config.json`.
  - `-log_file TEXT`: Specifies the path to the log file where operations details are stored.
  - `-metadata_path TEXT`: Path to the metadata file. This option is required.
  - `-metadata_type [real|synth]`: Specifies the type of metadata, where `real` stands for actual metadata and `synth` for synthetic. This option is required.
  - `-add_domain_knowledge`: Includes domain-specific knowledge in the query processing.
  - `-add_workflow_tags`: Includes workflow tags in the query to enhance context.
  - `-help`: Displays the help message and exits, showing all available command options.

## E.2  EVALUATION

Explain about evaluation in a line and then explain the CLI usage here.

The command `discovery_eval.py` is used to evaluate the outputs generated by the discovery agent. Below are the detailed descriptions of the command options:

- **Usage:** `discovery_eval.py [OPTIONS] QUERY` – Executes the evaluation agent with specified options and a query.
- **Options:**

- `-gold_hypo TEXT`: Specifies the gold standard hypothesis for comparison. This field is required.
- `-gold_workflow TEXT`: Specifies the gold standard workflow to be used as a reference during evaluation.
- `-pred_hypo TEXT`: Specifies the predicted hypothesis generated by the discovery agent. This field is required.
- `-pred_workflow TEXT`: Specifies the predicted workflow generated by the discovery agent.
- `-metadata_path TEXT`: Specifies the path to the metadata file that is utilized during evaluation. This field is required.
- `-metadata_type [real|synth]`: Determines the type of metadata used in the evaluation, where `real` indicates actual metadata and `synth` indicates synthetic metadata. This field is required.
- `-eval_output_path TEXT`: Specifies where the evaluation results should be saved.
- `-help`: Displays the help message and exits, detailing all available command options.

## F    EVALUATOR PROMPTS

We provide below the exact prompts used for our GPT-4 based evaluation of the generated hypothesis against the gold hypothesis.

**Listing 1** Decomposition Prompt to obtain context from a hypothesis.

```
decomposition_prompt = f"""\
        Given a set of dataset columns, a ground-truth hypothesis, and
        the analysis workflow used, your task is to extract the context
        present in the hypothesis.

        Here are the definitions for context:

        - Contexts: Boundary conditions that limit the scope of a
        sub-hypothesis. E.g., "for men over the age of 30", "in Asia and
        Europe", or "None" if there is no boundary condition specified.

        Make sure to only use the information present in the hypothesis
        and the workflow. Do not add any new information.
        If no context can be extracted, return an empty list.

        Here is the metadata for the task:
        ```json
        {{
            "datasets": {dataset_metadata},
            "hypothesis": "{hypothesis}",
            "workflow": "{workflow}"
        }}
        ```

        Return your answer as a JSON object in the following format:
        ```json
        {{
        "hypothesis":
            {{
                "text": the hypothesis in natural language,
                "context": a short text description of the context of
                the hypothesis,
                "explanation": a short text explanation for the
                breakdown of the sub-hypothesis
            }}
        }}```
        """
```

**Listing 2** Matching prompt to match contexts of two sub-hypotheses.

```
matching_prompt = f"""
        Given a gold hypothesis, a gold context, a predicted hypothesis,
        and a predicted context, your task is
        to determine if the predicted context semantically matches the
        ground-truth context.

        Here is the definition for Context: Boundary conditions that
        limit the scope of a sub-hypothesis. E.g., "for men over the age
        of 30", "in Asia and Europe", or "None" if there is no boundary
        condition specified.

        If the predicted context matches the gold context, return true,
        otherwise return false.

        Here is the metadata for the task:
        ```json
        {{
            "gold_hypothesis": "{gold_hypotheis}",
            "gold_context": "{gold_context}",
            "predicted_hypothesis": "{pred_hypothesis}",
            "predicted_context": "{pred_context}"
        }}
        ```

        Return your answer as a JSON object in the following format:
        ```json
        {{
            "match": true or false
        }}
        ```"""
```

**Listing 3** Prompt for variable alignment between two sub-hypotheses.

```
main_context = f"""
        You are going to compare two natural-language hypotheses HypoA
        and HypoB accompanied with optional workflows: WorkflowA for
        HypoA and WorkflowB for HypoB.
        Both the hypotheses answer the natural language query "QUERY"
        over the dataset(s) described by dataset description(s) and
        column description(s) below.
        Compare HypoA and HypoB in terms of three aspects: Contexts,
        Variables, and Relations.
        E.g., for the hypothesis "From 1995 to 2009, the number of
        sandhill cranes around the tundra (Indigilka River) surged by an
        astounding ~10X":
        * Contexts refer to the stratification of the data under which
        the given hypothesis is True. E.g., "For all women", "From 1995
        to 2009".
        * Variables refer to the set of variables (either dependent or
        independent) that are mentioned in the hypothesis. E.g., number
        of sandhill cranes, location.
        * Relations refer to the form of relation between the variables.
        E.g., "surged by ~10x".

        Answer the following questions for a given pair of hypotheses,
        HypoA and HypoB, along with an explanation grounded on the QUERY
        and the DATASET(S).

        Here is the metadata for the task:
        ```json
        {{
        "datasets": {datasets_json},
        "query": {query},
        "HypoA": {gold_hypo},
        "WorkflowA": {gold_workflow},
        "HypoB": {gen_hypo},
        "WorkflowB": {gen_workflow}
        }}
        ```

        {variable_question}"""
variable_question = """\
        Question: For both HypoA and HypoB, what are the different
        variables found in the hypotheses? \
        Return your answer as a JSON object in the following format:
        ```json
        {{
        "sizeA": num of variables used in HypoA
        "sizeB": num of variables used in HypoB
        "intersection": num of variables common in HypoA and HypoB. Use
        *fuzzy matching* to determine intersection, accounting for
        paraphrases or slightly different surface forms
        "explanation": a short text explanation about the variables
        }}```
        Answer:"""
```

**Listing 4** Prompt for relationship alignment between two sub-hypotheses.

```
main_context = f"""
        You are going to compare two natural-language hypotheses HypoA
        and HypoB accompanied with optional workflows: WorkflowA for
        HypoA and WorkflowB for HypoB.
        Both the hypotheses answer the natural language query "QUERY"
        over the dataset(s) described by dataset description(s) and
        column description(s) below.
        Compare HypoA and HypoB in terms of three aspects: Contexts,
        Variables, and Relations.
        E.g., for the hypothesis "From 1995 to 2009, the number of
        sandhill cranes around the tundra (Indigilka River) surged by an
        astounding ~10X":
        * Contexts refer to the stratification of the data under which
        the given hypothesis is True. E.g., "For all women", "From 1995
        to 2009".
        * Variables refer to the set of variables (either dependent or
        independent) that are mentioned in the hypothesis. E.g., number
        of sandhill cranes, location.
        * Relations refer to the form of relation between the variables.
        E.g., "surged by ~10x".

        Answer the following questions for a given pair of hypotheses,
        HypoA and HypoB, along with an explanation grounded on the QUERY
        and the DATASET(S).

        Here is the metadata for the task:
        ```json
        {{
        "datasets": {datasets_json},
        "query": {query},
        "HypoA": {gold_hypo},
        "WorkflowA": {gold_workflow},
        "HypoB": {gen_hypo},
        "WorkflowB": {gen_workflow}
        }}
        ```

        {variable_question}"""
dimension_question = """
        Question: Does HypoB exhibit the same relation as HypoA?
        Compare using the following example hierarchy of relationships
        (based on specificity): \
        "there exists a relationship" > "positive relationship" >
        "positive AND (linear OR quadratic)" > "positive AND linear."
        Options: A) very similar B) similar but general than HypoA C)
        different
        Return your answer as a JSON object in the following format:
        ```json
        {{
        "answer": one of the options from A) very similar B) similar but
        general than HypoA C) different
        "explanation": a short text explanation about the relationship
        comparison
        }}```
        Answer:"""
```

