# OpenReview forum: "DiscoveryBench: Towards Data-Driven Discovery with Large Language Models"
_ICLR.cc/2025/Conference — ICLR 2025 Poster_

### Official Review · Reviewer_rQhh · 2024-11-01

**Soundness:** 2
**Presentation:** 2
**Contribution:** 3
**Rating:** 5
**Confidence:** 4

**Summary:**

The work deals with the problem of data-driven discovery using large language models.

**Strengths:**

The authors attempt to formalize the problem of data-driven discovery, which, in my opinion, is a crucial step towards solving this task. In addition, a benchmark is introduced -- unfortunately, I did not have the chance to look at it in more detail, but the authors claim that they will publicly release it.

**Weaknesses:**

I am confused with the formulation of the problem of data-driven discovery as in Section 3. In line 049, the authors give an example of a data-driven discovery task: “How did urban land use affect the invasion of introduced plants in Catalonia?” The answer to this task is the ways urban land used affected the invasion of introduced plants in Catalonia (if this is the case). However, in Section 3, line 152, you define a hypothesis as a tuple of three elements and in line 141, you say that a hypothesis maps to supported or unsupported. So, if a hypothesis maps to true or false (supported or unsupported), then the example task that you have in line 049 cannot map to your formulation. Please clarify what is the case and, please specify concretely how the task in line 049 translates to your formulation.

I have a few other comments/questions regarding the formulation in Section 3:
- In line 139, the authors say that a hypothesis is a sentence/semantic tree, and then in line 152 the authors define a hypothesis as a ternary tuple. Please fix this inconsistency, as it is confusing to the readers.
- In line 141, the authors talk about a verification procedure \mathcal{V}_D mapping a hypothesis to supported and unsupported. Is my understanding correct, that the goal of the data-driven task is both to find whether a hypothesis is supported and unsupported and to return the exact \mathcal{V}_D? If my understanding is correct, then how do you define the space of valid verification procedures \mathcal{V}_D? I would kindly ask the authors to clarify this part, as it is missing.
- In line 152, the authors write h := \psi(c,u,r). That formulation essentially means that h (the hypothesis) is defined as its answer, \psi(c,u,r). I believe what the authors want to say is that \mathcal{V}_D(h) = \psi(c,u,r), i.e., that the answer to h is defined as \psi(c,u,r). Please fix the notation accordingly or clarify what you meant.
- Continuing with my previous comment, I would kindly ask the authors clarify what is the relationship between \mathcal{V}_D and \psi.
- In line 161, the authors give another definition of the data-discovery task: that of given a goal G, to find a hypothesis h. From Section 3, I understood that the objective is: given a hypothesis h = (c,u,r) to derive \psi (or \mathcal{V}_D). I would kindly ask the authors to define concretely the problem they are dealing with, addressing all my comments from above, as now it is difficult for the reader to understand what is happening.
- In line 293, the authors talk about hypothesis semantic trees, giving an example in Figure 1. However, in line 152, the hypothesis is a ternary tuple.
- In line 240, the authors talk about the implementation workflow, however, they give no definition/specification of them in Section 3.

The above issues, make it difficult for the reader to understand this work and the proposed benchmark. I am willing, however, to increase my score if the authors address my comments/questions.

**Questions:**

Please read my comments/questions from the above field. In addition:

- One suggestion that I have for the authors is to start with the example given in lines 247 and show how G, h, \psi, c, u, r, and \mathcal{V}_D look like for this example.
- Please clarify whether a hypothesis is represented as a sentence, a semantic tree, or a ternary tuple, and to explain how these different representations relate to each other if multiple are used.
- Please define the space of valid verification procedures V_D and clarify whether finding V_D itself is part of the task or if it's given.
- Another suggestion that I have for the authors is to look into automated theorem proving, (e.g., “Handbook of Automated Reasoning”, Volume 1, 2001), where the notions of hypotheses, theorem proving, proof trees, are formally defined. I would suggest using this notation to formally define the data-driven discovery task. I believe that this work will benefit a lot from a more rigorous formulation, e.g., to my understanding, the elements in \mathcal{V}_D map to proof trees.

**Details Of Ethics Concerns:**

There are concerns regarding data safety/privacy/proprietary data -- these issues arise with any benchmark that is released to the public as this one. Can the authors please quickly comment on these issues, e.g., if it is safe to release their benchmark and what steps have they taken to ensure that the benchmark complies with data protection schemes.

---

> ### Author Response · Authors · 2024-11-22
> **Authors' Response (1/3)**
>
> Thank you for taking the time to review our work and listing thoughtful comments and questions. We apologize if our expositions were not clear enough in some parts, and we try to clarify all of them in the rebuttal. We also appreciate your acknowledgment of the importance of designing a formal view of the data-driven discovery problem, as you did in this work. We updated our manuscript to address your comments and suggestions; all changes are denoted in red.
>
> __`W1: I am confused with the formulation of the problem of data-driven discovery as in Section 3 … “How did urban land use affect the invasion of introduced plants in Catalonia?” The answer to this task is the ways urban land used affected the invasion of introduced plants in Catalonia (if this is the case). However, … you define a hypothesis as a tuple of three elements and … say that a hypothesis maps to supported or unsupported. … Please clarify …`__
>
> Thank you for reading through the paper so thoroughly, and we apologize for the confusion with the formalism. We will try to clarify here.
>
> The discovery **goal** (or query) is *"How did urban land use affect the invasion of introduced plants in Catalonia?"*. Note that a goal is different from a **hypothesis**.
>
> The answer to a goal is a supported hypothesis (a sentence), e.g., *"Urban land use increased invasion by agriforest plants over gardening-introduced ones in Catalonia."* While listing the different ways in which urban land use affected invasion would be reasonable from a QA perspective, the expected answer in our setup (for data-driven discovery) is to capture a quantifiable relationship between variables of interest available in the dataset. Note also that the answer is the (supported) hypothesis, not the “supported” / ”unsupported” label on that hypothesis. Note, too, that a hypothesis is a sentence, not a triple, as we describe below.
>
> We updated Section 3 in the updated manuscript with this example from Figure 1.
>
> __`W2: …the authors say that a hypothesis is a sentence/semantic tree, and then … the authors define a hypothesis as a ternary tuple…`__
>
> Sorry for the confusion! To clarify, a hypothesis is a sentence. That sentence describes a context, the variables, and the relationships between those variables in that context. Conceptually, it is useful to think of the sentence as generated from these three elements by the generator function $\psi(c,v,r)$, but note the hypothesis itself is a sentence, not a tuple.
>
> __`W3: …Is my understanding correct, that the goal of the data-driven task is both to find whether a hypothesis is supported and unsupported and to return the exact \mathcal{V}_D? … how do you define the space of valid verification procedures \mathcal{V}_D? ...`__
>
> You are close! The goal is, in fact, to find a hypothesis h that answers the discovery goal (query) and that is supported by the dataset using some verification procedure $\mathcal{V}_D$, which we restrict to being a Python program that implements data processing and statistical modeling steps. Note that h is not provided and must be discovered. In this process, the agent often also discovers some $\mathcal{V}_D$. While our definition does not require $\mathcal{V}_D$ to be explicitly returned by the discovery agent, we do ingest that information, if returned, to improve the accuracy of LLM-based automatic evaluation.
>
> The space of valid verification procedures $\mathcal{V}_D$ is potentially any executable Python program. However, in our benchmark, the target (gold) hypotheses are intentionally designed to be very specific, making it highly improbable to arrive at them by chance or via shortcuts. The very low evaluation scores (Figure 4) substantiate this – even the best models do badly, suggesting they can't guess or take shortcuts to the target result.
>
> We clarified this in Line 141 in the updated transcript.
>
> __`W4: In line 152, the authors write h := \psi(c,u,r) ... I believe what the authors want to say is that \mathcal{V}_D(h) = \psi(c,u,r) ...`__
>
> __`W5: … clarify what is the relationship between \mathcal{V}_D and \psi.`__
>
> __`W6: … another definition of the data-discovery task: that of given a goal $G$, to find a hypothesis $h$. … I understood that the objective is: given a hypothesis h = (c,u,r) to derive \psi (or \mathcal{V}_D) ...`__
>
> It’s helpful to think of $\psi$ as a sentence generation function that takes $c$, $v$, and $r$ as arguments and generates a natural language sentence, namely the hypothesis $h$. Thus $h = \psi(c,v,r)$. Furthermore, $\mathcal{V}_D(h)$ determines whether $h$ is supported/unsupported, so the result of $\mathcal{V}_D(h)$ is one of {“supported”,”unsupported”}. E.g., $h$ = “there is a positive relationship between BMI and completion of the bachelor degree,” and $\mathcal{V}_D(h)$ = “unsupported”, indicating that the hypothesis $h$ is not supported by data $D$.

---

> ### Author Response · Authors · 2024-11-22
> **Authors' Response (2/3)**
>
> (Continued)
>
> __`W7: …the authors talk about hypothesis semantic trees, giving an example in Figure 1. However, in line 152, the hypothesis is a ternary tuple.`__
>
> Again, sorry for this confusion. To clarify, a hypothesis is a sentence that can be generated from $c$, $v$, $r$ by a generator function $\psi(c,v,r)$.
>
> A hypothesis semantic tree (Fig. 3) is not a hypothesis itself but a tree of variables depicting the dependencies between the variables in the dataset (leaf nodes), intermediate variables, and the target (dependent) variable of the hypothesis (root node). As a simplified example, consider the hypothesis “Health depends on BMI and age.”. Here, the target variable (health) is set as the root node, and independent variables (BMI and age) are set as its children. BMI and age may themselves be considered as a target variable of a sub-hypothesis. E.g., “BMI is a function of height and weight.” Here, we add height and weight as children to the BMI node, thus making BMI an intermediate node in the tree. This recursive splitting can be done until the variables present in the dataset are reached, forming the leaves of the tree.
>
> We have clarified this with the updated text on Line 296.
>
> __`W8: In line 240, the authors talk about the implementation workflow; however, they give no definition/specification of them in Section 3.`__
>
> Implementation workflows are the description of the implementation efforts that we followed while reproducing the results from the original work to create the Real-DB benchmark. We curate a natural language version of the implementations and include it as metadata for each task. If the discovery agent returns a workflow (which is optional), we ingest the implementation workflow as an additional guide to improve the accuracy of LLM-based automatic evaluation of h.
>
> We have added this definition as a footnote on Line 229 in the updated draft.
>
> __`Q1: One suggestion … start with the example given in lines 247 and show how G, h, \psi, c, u, r, and \mathcal{V}_D look like for this example.`__
>
> Certainly! Here is the breakdown of the example:
> $G$ =  “How does socioeconomic status impact college degree completion for females compared to males?”; $h$ = “The effect of socioeconomic status on college degree completion is higher for females (0.4995) than males (0.4467)”; $\psi$ is a fixed sentence generator function that generates $h$ from $c$, $v$ and $r$. Here $c$ = "for gender groups male and female", $v$ = {socioeconomic_status, degree_completion}, $r$ = "higher for females (0.4995) than males (0.4467)"}. $\mathcal{V}_D$ = <Python code for linear regression>.
>
> __`> Q2:  Please clarify whether a hypothesis is represented as a sentence, a semantic tree, or a ternary tuple …`__
>
> From the answer to W2, a hypothesis is a sentence (not a tuple) describing a context, the variables, and the relationships between those variables in that context. Conceptually, it is useful to think of the sentence as generated from these three elements by the generator function $\psi(c,v,r)$.
>
> __`> Q3: Please define the space of valid verification procedures \mathcal{V}_D and clarify whether finding \mathcal{V}_D itself is part of the task or if it's given.`__
>
> The verification procedure $\mathcal{V}_D$ is not given. Hypothesis h must be discovered, and, in this process, $\mathcal{V}_D$ is often discovered as well, but is not part of our task. We, however, do ingest that information, if returned by the agent, to improve the accuracy of LLM-based automatic evaluation of $h$. Summarizing from the answer to W3, our target hypotheses are intentionally designed to be very specific, making it improbable to arrive at them by chance. We, therefore, do not impose a restriction on the space of valid Python programs.
>
> __`> Q4: Another suggestion … is to look into automated theorem proving … to formally define the data-driven discovery task. I believe that this work will benefit a lot from a more rigorous formulation, e.g., to my understanding, the elements in \mathcal{V}_D map to proof trees.`__
>
> This is a great suggestion! Proof trees in automated theorem proving indeed have a clear connection to our definition of $\mathcal{V}_D$ in data-driven discovery. However, an important difference between theorem proving and data-driven discovery is that in the former, the objective is to find the verification procedure given a theorem stated upfront, whereas the objective in our setup is to find the hypothesis itself that is verifiable, i.e., verification forms a sub-routine in data-driven discovery and is not the actual sole objective. Finally, we only evaluate the h with respect to the gold (target) hypothesis, irrespective of its $\mathcal{V}_D$.

---

> > ### Author Response · Authors · 2024-11-22
> > **Authors' Response (3/3)**
> >
> > __`Ethics concern: ... what steps have they taken to ensure that the benchmark complies with data protection schemes`__
> >
> > All NLS-related datasets are licensed under CCZero, and the rest of the datasets are licensed under CC by 4.0, which allows them to be freely used/distributed as long as they are properly cited. In NLS and ML engineering datasets, where there are human subjects, datasets are anonymized to protect respondent confidentiality.
> >
> > We additionally added this note in the appendix with our existing comment about the dataset licenses (Line 793).

---

> > > ### Author Response · Authors · 2024-11-25
> > > **Follow up: Concerns addressed?**
> > >
> > > Just to follow up: We hope we have fully addressed your concerns in our responses; please let us know, or if not please don't hesitate to let us know anything that is still outstanding so we can follow up with you. Thanks!

---

### Official Review · Reviewer_9u8G · 2024-11-03

**Soundness:** 4
**Presentation:** 3
**Contribution:** 3
**Rating:** 8
**Confidence:** 4

**Summary:**

The advances in data science and AI seem to be evolving the process of scientific discovery (the scientific method) by (drastically) reducing the time needed to formulate hypotheses, conduct experiments, and evaluate results. This paper proposes a benchmark to evaluate the potential role of LLMs in the automatic formulation of hypotheses, starting from a dataset and a natural language question about that dataset.

Typically, for a classification or regression task, a human will identify certain attributes of interest (A_1, …, A_n), a target attribute Y, and attempt to build a relationship between these elements (A_1, …, A_n → Y). This is the current process for generating new knowledge. In this work, the authors aim to determine whether using LLMs allows for the production of valid knowledge in a more general setting: providing a dataset, posing a general question in natural language to the system using an LLM, and obtaining a response in the form of a scientific discovery. The main question the authors seek to answer is how well this approach performs using state-of-the-art LLMs.

There are some recent tools that evaluate the performance of LLMs in testing well-defined hypotheses (clearly formulated questions on a dataset). These methods utilize a constrained search space to look for hypotheses. DiscoveryBench expands the search space by finding hypotheses in a broader context, closer to a human approach. As expected, the performance of LLMs decreases in this setting. Indeed, the paper shows that the best results achieve 25% performance (which I interpret as 25% of responses being valid knowledge, even if it is a bit more complex to interpret).

**Strengths:**

- The role of LLMs in the scientific method is still unknown (if one exists), and evaluating their capacity to accelerate the process of knowledge discovery is a topic of significant interest. This work represents an incremental advancement in this domain, providing a formal definition of data-driven hypotheses and expanding the search space for these hypotheses. The paper also proposes evaluating proposed hypotheses against a gold standard using semantic similarity measures. In my opinion, this work is of genuine interest to the community.

- The proposition is well described, and the code is available on GitHub. The authors have done impressive work collecting datasets, defining objectives for these datasets, and constructing gold standard hypotheses.

- The results are clearly presented and discussed, demonstrating the limitations of current LLMs. This methodology can be utilized in the future to observe potential progress in the field of LLMs.

**Weaknesses:**

- The process of finding a data-driven hypothesis can be time/energy-consuming. This aspect is not discussed in the paper, and I understand that the page limit might not allow space for such discussions. Although this is not the main focus of the paper, it could be useful to explore whether there is a relationship between the size of the search space and the performance of the results.

- The proposed formalism for defining data-driven hypotheses, discovery goals, and task difficulty is somewhat limited. At the same time, I understand that human expertise is essential in this process, and complete automation is difficult, if not impossible, to achieve.

**Questions:**

How can it be ensured that the data alterations in the proposed datasets (page 6) do not lead to contradictions with the discovery goal, resulting in only one answer—the target hypothesis?

Minor remarks:
 - Pg 3 : discopagevery --> discovery
 - Pg 6: Sec ?? --> update the number of the section

---

> ### Author Response · Authors · 2024-11-22
> **Authors' Response**
>
> Thank you for taking the time to review. We are thankful for your encouragement on our benchmark for its potential wider acceptability, thoughtful curation, and clear and insightful evaluation. We address all your comments and make necessary edits in the manuscript to reflect the changes in red.
>
> __`W1: it could be useful to explore whether there is a relationship between the size of the search space and the performance of the results.`__
>
> Thank you for your suggestion. We tried to explain this in Section 4.1. For most real datasets, we have incomplete a priori information about unobserved intermediate variables and their association with observed ones. Hence, the exact estimation of the search space size is often impractical to compute. To get an estimate, we approximate the size of the search space with task difficulty, measured as the length of a successful implementation workflow required to derive a target hypothesis.
>
> We derive two types of tasks: easy and hard, from the same hypothesis, where for easy, we provide the derived variables as observed variables in the dataset (e.g., BMI), and for hard, it would require deriving intermediate variables (BMI from height and weight) to reach the
> target. Intuitively, discovery becomes harder as the hypothesis search space increases (Lines 244-246 in the updated manuscript).
>
> Figure 5c shows that discovery agents perform better on easier tasks than on harder ones, where easier tasks entail a smaller search space, while the harder ones span a much larger search space.
>
> __`W2: The proposed formalism for defining data-driven hypotheses, discovery goals, and task difficulty is somewhat limited. At the same time, I understand that human expertise is essential in this process and complete automation is difficult, if not impossible, to achieve.`__
>
> Right, our formalism treats a hypothesis in terms of its <context, variables, relations>. While this is not perfect, it actually covers a very large space of hypotheses (we did not encounter any when building DiscoveryBench that could not be mapped to this way of thinking), especially given that the relations can be arbitrarily complex. So we did not find it too limiting, although clearly, it would not cover everything.
>
> Furthermore, we completely agree that human expertise is essential for the scientific discovery process. The purpose of our formalism and benchmark is to evaluate the extent to which current automated methods can assist in the discovery process.
>
> __`Q1. How can it be ensured that the data alterations in the proposed datasets (page 6) do not lead to contradictions with the discovery goal, resulting in only one answer—the target hypothesis?`__
>
> The perturbations we add during data generation simply mimic the noise in real-world data collection. In particular, for each column, we add Gaussian noise scaled to only 5% of the standard deviation in values generated for that column and clip the perturbed values to the original minimum and maximum, thus keeping the range consistent with the LLM-provided specification. Thus, our perturbations are minimal and do not alter the true relationship described by the data-generating pandas expression.
>
> __`Minor remark: Fix typos.`__
>
> Thank you for spotting these. We fixed them in the updated manuscript.

---

### Official Review · Reviewer_8cxn · 2024-11-03

**Soundness:** 3
**Presentation:** 3
**Contribution:** 4
**Rating:** 8
**Confidence:** 3

**Summary:**

In this paper, the authors consider the question of how capable
state-of-the-art LLMs are at automated data-driven discovery. More
precisely, the authors present a benchmark for this task, designed not
only to assess LLMs’ capability in discovery tasks but also to provide
information for improving these capabilities.

The benchmark presented in this paper consists of 264 manually
collected tasks, along with 903 tasks that were synthetically
generated. This benchmark is used to evaluate several LLM-based
reasoning frameworks, providing useful information on the current
capabilities of such systems in automated data-driven discovery.

**Strengths:**

S1) The authors introduce a simple yet expressive notion of
data-driven hypothesis. Discovering and validating such hypotheses
from a dataset is a challenging problem for LLMs.

S2) The authors develop a comprehensive benchmark to test the
capabilities of LLMs in discovering data-driven hypotheses.

S3) The authors use the benchmark to test some popular LLM-based
reasoning frameworks, drawing useful conclusions about the
capabilities of these systems in discovering data-driven hypotheses.

**Weaknesses:**

W1) As part of the benchmark, the authors developed some synthetic
tests. These tests are supposed to capture synthetic task examples
constructed from workflows in published works. However, the authors do
not clearly explain in what sense these synthetic tests properly
represent these workflows.

**Questions:**

The generation DB-SYNTH consists of four steps intended to capture
synthetic task examples from workflows in published works. In
particular, a task dataset is constructed in the third step by
generating synthetic data using various sampling strategies. The
authors do not explain how these steps, and particularly the data
generation step, provide good representations of the task examples from
workflows in published work (notice that there is a missing reference
to a section in the data generation paragraph). This justification
should be part of this work, as if the synthetic tasks are not good
representatives of the real examples, then we could be asking the LLMs
to solve tasks that may be too simple or too complicated, which could
reduce the quality of the tasks in the benchmark.

---

> ### Author Response · Authors · 2024-11-22
> **Authors' Response**
>
> Thank you for your thoughtful review. We appreciate your positive comments on the expressivity, comprehensiveness, and evaluation results of our benchmark. We address your questions and make necessary edits in the manuscript to reflect the changes in red.
>
> __`W1: … the authors developed some synthetic tests. … However, the authors do not clearly explain in what sense these synthetic tests properly represent these workflows.`__
>
> __`Q1: DB-SYNTH: The authors do not explain how these steps, and particularly the data generation step, provide good representations of the task examples from workflows in published work (notice that there is a missing reference to a section in the data generation paragraph). This justification should be part of this work …`__
>
> Thanks for catching the missing reference, which actually leads to the detailed discussion of the procedure in the Appendix. We have added the reference in the updated manuscript.
>
> To construct the semantic tree, for leaves, we use different sampling strategies based on the data type. Specifically, for categorical nodes, we sample instances with replacement from a range of allowed values generated by an LLM, whereas for numeric, we first select a distribution (e.g., normal) and its parameters based on the specified range, again generated by the LLM, and then perform sampling. For each subsequent level in F, we create new columns for nodes by executing the pandas expressions associated with the generated hypothesis.
>
> The expressions associated with the hypothesis themselves are generated by prompting the model with natural-language workflow steps, e.g., “within-cluster analysis” and “temporal analysis.” We first curate a pool of real workflow steps from DB-Real workflows. Then, to encourage diversity in plausible (according to the LLM) workflows and to increase coverage (by accounting for workflows that we could not cover in DB-Real for practical/reproducibility reasons), we allow LLMs to combine several real-workflow steps (type of variable derivation, type of relationship derivation, etc) plausibly through drawing from the real workflow pool. This significantly improves the likelihood of testing synthetic workflows that connect well with the real ones. Moreover, we also use self-refine and human filtering to monitor the quality of the expressions and the variable metadata therein after tree and data generation. E.g., we saw a substantial improvement in quality when generating variable values from a specified range rather than a random range.
>
> We added this additional justification at Line 304 and Appendix C in the updated manuscript.

---

### Author Response · Authors · 2024-11-22
**Authors' General Response to Reviewers (Summary of Manuscript Changes)**

We sincerely thank all the reviewers for their encouraging, thoughtful, and detailed comments on improving the manuscript. We have addressed all the comments and questions. Additionally, we have updated the manuscript to incorporate changes (__`marked in red`__) suggested by the reviewers to improve the readability of the paper. Here is a **summary of the manuscript changes**:

- Updated exposition of the data-driven discovery formalism, including the definition, scope, and example (addressing [Reviewer rQhh](https://openreview.net/forum?id=vyflgpwfJW&noteId=LVZQ6yB71m))
- Added the definition of  implementation workflow (addressing [Reviewer rQhh](https://openreview.net/forum?id=vyflgpwfJW&noteId=tgjOJr9dH5))
- Updated discussion on hypothesis semantic tree (addressing [Reviewer rQhh](https://openreview.net/forum?id=vyflgpwfJW&noteId=tgjOJr9dH5))
- Fixed the missing section reference and typos (addressing [Reviewer 9u8G](https://openreview.net/forum?id=vyflgpwfJW&noteId=aQVnrwkJCB))
- Updated discussion on synthetic data generation (addressing [Reviewer 8cxn](https://openreview.net/forum?id=vyflgpwfJW&noteId=XnhAReXY0o))
- Updated dataset licenses (addressing [Reviewer rQhh](https://openreview.net/forum?id=vyflgpwfJW&noteId=X780jocKIH))

We’d be happy to address any additional comments/questions!

---

### Meta-Review · Area_Chair_14t4 · 2024-12-19

**Metareview:**

This paper introduces a novel approach to evaluating large language models (LLMs) by leveraging them for data discovery. Two of the reviewers view the paper positively, recognizing it as an important and innovative method for LLM evaluation. However, the third reviewer believes the paper is not yet ready for publication, citing a need for greater clarity to ensure its effectiveness.

Given the rapid advancements in the field, delaying this publication may not be ideal. I recommend that the authors focus on improving the clarity of their explanations, particularly regarding how the proposed comparison framework operates. It is essential for non-LLM specialists to easily understand the framework's purpose and contributions.

**Additional Comments On Reviewer Discussion:**

There was engagement between the reviewers and authors.

---

### Decision · Program_Chairs · 2025-01-22

Accept (Poster)